# Cytokines and Venous Leg Ulcer Healing—A Systematic Review

**DOI:** 10.3390/ijms23126526

**Published:** 2022-06-10

**Authors:** Ewa A. Burian, Lubna Sabah, Tonny Karlsmark, Klaus Kirketerp-Møller, Christine J. Moffatt, Jacob P. Thyssen, Magnus S. Ågren

**Affiliations:** 1Department of Dermato-Venereology & Wound Healing Center, Bispebjerg Hospital, University of Copenhagen, 2400 Copenhagen, Denmark; ewa.anna.burian@regionh.dk (E.A.B.); lubna.sabah@regionh.dk (L.S.); tonny.karlsmark@regionh.dk (T.K.); klaus.kirketerp-moeller.01@regionh.dk (K.K.-M.); christine.moffatt@virgin.net (C.J.M.); jacob.pontoppidan.thyssen@regionh.dk (J.P.T.); 2Centre for Research and Implementation of Clinical Practice, London W5 4QB, UK; 3Nottingham University Hospitals, NHS Trust, Nottingham NG7 2UH, UK; 4Digestive Disease Center, Bispebjerg Hospital, University of Copenhagen, 2400 Copenhagen, Denmark; 5Department of Clinical Medicine, Faculty of Health and Medical Sciences, University of Copenhagen, 2200 Copenhagen, Denmark

**Keywords:** chronic wound, wound healing, cytokine, growth factor, inflammation, biomarker

## Abstract

Venous leg ulcers (VLUs) are the most common type of leg ulcers with a significant socioeconomic burden due to slow healing. Cytokines may be involved in the pathogenesis of VLUs. In this systematic review, our objective was to investigate the association between cytokine levels, including growth factors, with the healing of VLUs. PubMed, Embase, Web of Science and Cochrane Library were searched from their inception to August 2021. We retrieved 28 articles investigating 38 different cytokines in 790 patients. Cytokines were most commonly investigated in wound fluid and less frequently in biopsies and serum. The studies were judged as having a moderate to high risk of bias, and the results were often inconsistent and sometimes conflicting. A meta-analysis was not performed due to clinical and methodological heterogeneities. We found weak evidence for elevated IL-1α, IL-6, IL-8, TNF-α and VEGF levels in non-healing VLUs, an elevation that declined with healing. TGF-β1 levels tended to increase with VLU healing. Other cytokines warranting further investigations include EGF, FGF-2, GM-CSF, IL-1β, IL-1Ra and PDGF-AA/PDGF-BB. We conclude that non-healing VLUs may be associated with an elevation of a palette of pro-inflammatory cytokines, possibly reflecting activated innate immunity in these wounds. There is a paucity of reliable longitudinal studies monitoring the dynamic changes in cytokine levels during wound healing.

## 1. Introduction

Venous leg ulcers (VLUs) are the most common type of leg ulcers [1]. The annual prevalence is above 1.5% among the elderly [2], a number that is increasing [3]. Treatment of patients with non-healing wounds excavates 2.5% of the healthcare budget [4,5] and can lead to a negative impact on the patient’s quality of life [6]. Non-healing wounds do not follow the timely sequence of normal repair, which often results in prolonged healing [7,8,9]. Despite optimal compression therapy [10,11,12], about 25% of VLU patients are unlikely to achieve healing within six months [13,14]. Comorbidities may impact healing in VLU patients [3]. Increased understanding of the factors that contribute to non-healing VLUs could improve treatment outcomes. One part of unravelling this complex puzzle of factors may involve understanding the role of cytokines.

Cytokines are a large family of small proteins (~10–30 kDa) produced by virtually all cells in the body, including immune cells [15,16]. Over 100 genes code for proteins with cytokine-like functions and have accurately been termed “a nomenclature minefield” [15,16,17,18]. In general, cytokine concentrations are low under normal physiological conditions but increase sharply in response to different stimuli [19,20]. By binding to receptors, cytokines activate intracellular signaling pathways eliciting downstream effects [16]. Cytokines regulate protein synthesis, migration, proliferation and differentiation of cells and have pluripotent functions, such as being potent mediators of inflammatory processes [20,21].

Wounds normally heal through the sequential but overlapping inflammatory, proliferative and remodeling phases. The complex process of wound healing involves interaction between multiple cell types and signaling molecules [9,22]. Most of our knowledge regarding the temporal and spatial variations of cytokines in wound healing derives from acute surgical wounds [23,24,25]. This knowledge has been supplemented with wound healing studies in knock-out mice models that have convincingly shown that deficiencies of cytokines severely impair wound healing [26,27,28,29], which underscores the importance of cytokines in normal wound healing.

The development of VLUs is thought to be initiated through venous hypertension, leading to the extravasation of inflammatory cells and the release of proteinases, causing skin breakdown and finally ulceration [30,31]. This has sparked research interest in the role of inflammation in the pathogenesis of non-healing wounds. It has been suggested that a pro-inflammatory wound environment prevents the healing of VLUs [32,33]. Differences in the composition of the immune cell infiltrate between acute wounds, and VLUs have implications on the cytokine network [34,35]. In contrast, other investigators have hypothesized that an initial temporary state of increased inflammation is important for VLUs to heal [36].

Harris et al. were the first to investigate the relationship between the healing state of VLUs and cytokine levels in 1995 [37]. Wound fluid was collected from 18 patients with either non-healing or healing VLUs. No significant differences in the levels of the examined cytokines were found. This pioneering work has been followed up by multiple research groups attempting to uncover the role of cytokines in VLU healing, but with divergent results.

To decipher these results, we performed the present systematic review of the literature to find evidence regarding the association of endogenous cytokines with the healing of VLUs. This was assessed by including studies comparing cytokine levels between non-healing and healing VLUs and by correlating cytokine levels to healing.

## 2. Materials and Methods

### 2.1. Study Types

We included studies providing sufficient data regarding quantitative or semi-quantitative levels of endogenous cytokines in patients with VLUs and associating the findings with healing. We included studies that had measured cytokines by any method. Case-series, case–control, cross-sectional and cohort studies with at least five patients were eligible for inclusion, including interventional studies, provided that the level of the endogenous cytokine was correlated to an outcome of interest. Single case reports, expert opinions, reviews, abstracts, conference presentations or guidelines were not included. Non-human, in vitro and ex vivo studies were excluded. Studies comparing VLUs to other wound types, healthy controls or healthy skin were not investigated unless the data for VLUs were presented separately and related to the outcome of interest. Non-English studies were excluded.

### 2.2. Target Population

Patients with open VLUs were included, for whom there were available published cytokine/growth factor levels. We accepted the authors’ definition of VLU regardless of the method of diagnosis used. Mixed venous-arterial ulcers were accepted. Studies reporting on patients with “leg ulcers” were included if evidence for venous insufficiency was reported and provided that at least 75% of the patients had VLUs. We included studies irrespective of the healing status (e.g., healing or non-healing) or the wound duration. We accepted the authors’ definitions of the status/outcome of the wound.

### 2.3. Cytokines Selected for the Search

We chose the following cytokines for investigation: EGF, FGF, G-CSF, GM-CSF, HGF, IFN-γ, IGF, IL-1 to IL-40, KGF, M-CSF, NGF, PDGF, S100A8/A9, TGF, TNF and VEGF, and their receptors. Growth factors and S100A8/A9 are in this review classified as cytokines. We included one classical chemokine, IL-8.

### 2.4. Search Strategy

The protocol for this review was published on the homepage of Prospero (Centre for Reviews and Dissemination, University of York, York, UK) in advance of our search and data extraction [38]. We initially planned to include studies investigating different local wound characteristics on the expression of cytokines, e.g., infected vs. uninfected VLU. Due to the large number of studies identified for our primary endpoint, we restricted ourselves to studies investigating healing (initiating compression therapy was equated with healing). The PRISMA flow chart was used for the search strategy and retrieval of records (Figure 1).

The databases PubMed (including MEDLINE), Embase (1974 to 17 August 2021), Web of Science and Cochrane Library were searched on 18 August 2021. Searches of “venous leg ulcers”, “cytokines”, “growth factors” and the name of each specific cytokine/growth factor were conducted, including an exhaustive list of terms relating to these terms. Free text and MeSH terms/keywords were applied. An experienced librarian at the University of Copenhagen scrutinized the final search strings. The search strings are found in Appendix A. No search filters were used. Studies retrieved from each database were imported into EndNote and uploaded to Covidence software (Melbourne, Australia), assisting in duplicate removal and facilitating the study selection process. The software allowed for independent screening of titles/abstracts by two authors (E.A.B. and L.S.) and identified conflicts. These were resolved by discussion. Studies that were clearly ineligible were excluded, while the others were retrieved in full text, according to the eligibility criteria. Reasons for exclusion were given. Disagreements were discussed and resolved by a third reviewer (M.S.Å.) when the agreement was not reached. Reference lists of relevant reviews were searched.

### 2.5. Data Collection and Extraction

Reliable data were extracted from the text, tables and figures of the articles by one reviewer (E.A.B.) and entered into a standard Excel data collection form designed a priori. The following information was entered: reference, year of publication, country, study design, eligibility criteria, sample size, outcomes related to the cytokines/growth factors, definition of outcome, type of cytokine/growth factor analyzed, duration and follow-up, study population, disease status at inclusion, treatment before and after inclusion, diagnosis of VLU, patient characteristics (age, sex, comorbidities, infection, ankle-brachial index, etc.), wound characteristics (e.g., wound age and size), interventions, characteristics of specimen collection, analysis, processing, storage, time-point of biomarker collection, missing data and causes, adjustments for confounders, results, statistics and conclusions. The results for each cytokine were extracted to another sheet aggregating results from the same cytokine under one heading—provided that a test of significance had been performed. Concentration or the correlation coefficient was extracted. The extracted data were checked by a second author (L.S.).

### 2.6. Risk of Bias (Quality) Assessment

Quality assessment was performed independently by two authors (E.A.B. and L.S.). Disagreements were resolved in consensus or with a third author (M.S.Å.). A critical assessment of the study quality was conducted using a modified version of the Quality in Prognostic Studies (QUIPS) tool [40]. This tool has been evaluated as appropriate for prognostic studies [40,41,42]. It consists of 31 questions divided into six domains, which are graded as low, moderate or high. The domains include study participation, study attrition, prognostic factor measurement, outcome measurement, study confounding and statistical analysis and reporting. As not all included studies could be regarded as prognostic factor studies (with the measurement of the biomarker at baseline to predict a subsequent outcome), the domain “prognostic factor” was interpreted as the risk of bias related to the method of the cytokine measurement. 

We performed an overall rating of each study, taking all domains into account. A low risk of bias in all domains was regarded as a study with an overall low risk of bias. A moderate risk of bias was judged if there was a high or moderate risk of bias for at least one domain. A high risk of bias was judged if there was a high risk of bias for at least two domains. In studies investigating multiple outcomes and/or cytokines, the risk of bias assessment reflects the overall judgment of the paper.

To minimize the risk of a subjective judgment and bias, we agreed on pre-specified criteria for each domain. We put emphasis on sufficient description of baseline data (age, ulcer duration, ulcer size, ankle-brachial pressure index (ABPI) and description of infection), diagnosis of VLU with confirmation by an imaging technique, such as Duplex, and risk of selection bias by including patients that were not representative of the whole review population of interest. As for attrition bias, a drop-out rate of a maximum of 20% was acceptable. A clear and appropriate methodology for each cytokine and outcome was assessed, including a clear description of the time-point of evaluation. Predefined key confounding factors were chosen based on a Cochrane review identifying wound size, wound duration, age and infection as potential prognostic factors for healing in VLU [42]. We also added the risk of confounding by the treatment initiated (other than standard care). Finally, the choice of statistical test was evaluated, including a clear and adequate data presentation, with a limited amount of selective reporting.

### 2.7. Measures of Association, Data Synthesis and Pooling of Data

Synthesizing data in these types of studies is challenging due to the different reporting of results, including the statistics used. We expected the data to mainly be presented as continuous data or as correlation analysis. We planned to perform a meta-analysis only if this was justified. Reviewing the studies, we deemed it inappropriate to pool data in the form of a meta-analysis due to the overall high risk of bias, large clinical and methodological heterogeneity (see results) and diverse presentations of results. Conversions of median values to means were also regarded as inappropriate. Therefore, we conducted a synthesis in the form of a summary of findings table(s) including a qualitative and narrative presentation of the findings, avoiding a fragmented and potentially misleading forest plot. For studies presenting the absolute concentration of cytokines between healing and non-healing VLUs, fold change was calculated. 

### 2.8. Missing Data

We included studies, regardless of whether there was missing data or limited evidence of the effect size, e.g., if the cytokine measured was “non-significant”. We did not contact the study authors to attempt to retrieve missing data or for a clarification of uncertainties due to the overall high risk of bias in the included studies. This would also have delayed the completion of the review.

## 3. Results

After the removal of duplicates of the 3515 records retrieved, 2546 titles and abstracts were screened (Figure 1). One hundred and six studies were read in full text, and 28 of these were found eligible for inclusion.

### 3.1. Characteristics of Included Studies and Participants

The 28 studies were published in English between 1995 and 2021 and were conducted in the US (*n* = 7), UK (*n* = 6), Italy (*n* = 5), Australia (*n* = 5), Ireland (*n* = 2), France (*n* = 1), Hungary (*n* = 1) and Poland (*n* = 1). Most of the studies were prospective (Table 1). A total of 790 patients were enrolled, with a median of 21 patients per study (range: 5–80 patients). Information regarding ankle-brachial index (ABPI) was present for 649 patients. Of these patients, 518 (80%) had VLU without any ischemia (ABPI < 0.8), while 20% allowed the inclusion of mixed venous-arterial ulcers. Venous insufficiency was confirmed by an imaging technique such as Duplex ultrasound or photoplethysmography in 19 of 28 studies. The remaining studies did not report how the diagnosis was made. Information regarding sex was available for 581 participants; among these, 313 (55%) were females. The overall mean/median age was approximately between 50–70 years. Of the 28 studies, eleven excluded patients with wound infection/cellulitis, five excluded diabetic patients and eleven studies excluded patients with autoimmune diseases. Patients receiving immunosuppressive treatments were excluded from ten studies. Twelve studies provided information regarding treatment prior to inclusion and were, in most cases, described as standard care and/or compression. We assumed that the number of wounds equaled the number of patients. Detailed data on patient characteristics for each study are found in Appendix A.

### 3.2. Collection of Specimen, Analysis, and Processing

Wound fluid was the most investigated specimen (*n* = 18 studies), followed by biopsy (*n* = 9), serum or plasma (*n* = 6), and one study investigated cytokines in peripheral blood mononuclear cells. 

Wound fluid was sampled using various methods, the most common being collection under a transparent occlusive dressing for 1–4 h. In one protocol [33], patients were given 1 L of water and then kept their legs dependent to increase wound fluid secretion. Other studies extracted wound fluid from gauze [54,55], dressings [47,58], PerioPaper collection strips [67], or from foam-tipped applicators [56]. Sampling duration of wound fluid ranged from 0 h to 1 week. Wound fluid samples were filtered in five studies [33,37,47,62,63]. Centrifugation of wound fluid prior to freezing was performed in 13 studies but not in four. In the majority of studies, wound fluid samples were stored at −80 °C until analyzed. 

Biopsies were taken from either the wound bed, wound edge or the perilesional skin for cytokine quantification, immunohistochemistry, real-time quantitative reverse transcription polymerase chain reaction and high throughput cDNA microarray.

Venous blood was collected from the arm or the leg from veins draining the ulcer area.

Cytokine concentrations were most often determined by enzyme-linked immunosorbent assay (ELISA) (in 61% of the studies), followed by multiplex assays (21%) and were most commonly expressed as pg/mL, while six studies normalized cytokine levels to the protein or albumin content [36,37,44,51,60,61]. No study expressed the cytokine levels to the wound size. Details of specimen collection and processing are found in Appendix A.

### 3.3. Definitions of Healing

The wounds were specified as non-healing, chronic or refractory by 82% of the studies (23 of 28) at inclusion, with diverse definitions (Appendix A). Wounds were typically >2 months old, with a mean/median wound size between 2.9 and 63.5 cm^2^. The definitions of healing also greatly varied, including the time-point of cytokine collection in the healing process (Appendix A). Both partial healing and complete healing were used as outcomes. Most studies used percentage area change with or without a cut-off over a time period ranging between 2 weeks and 13 months, e.g., 40% ulcer area reduction over 4 weeks. Only three studies had complete healing as an outcome [44,51,53]. Others assessed healing as a subjective clinical evaluation (“increased granulation tissue”) or did not provide a definition. Two studies compared inflammatory and granulating VLUs with each other [54,55], which we classified as non-healing and healing. The follow-up time of the included studies ranged from 0 to 13 months or until healing.

Dichotomous outcomes with healing vs. non-healing wounds or correlation analysis associating cytokine levels (or change in cytokine levels) and wound size change was most commonly assessed. 

### 3.4. Cytokines in Non-Healing/Healing VLUs

Thirty-eight different cytokines were investigated. Significant results and trends are presented in Table 2. The table includes studies comparing non-healing with healing VLUs and studies correlating cytokines to healing. The most studied cytokines in relation to healing were IL-8, TNF-α and VEGF, with at least ten different studies investigating each cytokine. Non-healing VLUs were reported to be associated with significantly elevated levels of IL-α, IL-6, IL-8, TNF-α and VEGF, which decreased with healing. On the contrary, increased TGF-β1 levels paralleled healing. These findings were supported by at least three publications per cytokine, reporting a significant (*p* < 0.05) result or trend (*p* < 0.10). IL-α, IL-8, TNF-α and VEGF increased in all three tissues: wound fluid, biopsies and blood. For IL-6 and TGF-β1, significant findings were found for wound fluid and biopsies but not for blood. Less consistent findings were found for EGF, FGF-2, GM-CSF, IL-1β, IL-1Ra and PDGF. Increased FGF-2 and PDGF were possibly linked to healing VLUs [54,58,64]. Findings from each study are shown in Appendix A. Appendix A presents a detailed extraction broken down for each cytokine.

Six studies reported absolute or relative levels of cytokines from repeated measurements over the course of VLU healing [47,56,59,62,63,67]. In general, cytokine levels fluctuated considerably over time, and the pattern differed among specific cytokines.

The association between baseline cytokine levels and subsequent healing was investigated. VEGF was significantly higher in baseline wound fluid of VLUs that had not healed compared to VLUs that had healed after 1 year [44]. Gohel et al. found no correlation between baseline wound fluid levels of FGF-2, IL-1β, TGF-β1, TNF-α or VEGF and healing at five weeks [48]. These findings agreed with others, where baseline IL-8, KGF or VEGF were not associated with healing at 12 weeks [61]. Mwaura et al. [58] reported increased baseline levels of PDGF-AA in wound fluid and biopsies at baseline in healing (>20% area reduction after eight weeks) compared to non-healing VLUs. Another study had a different approach and investigated whether cytokine change over two weeks could predict subsequent healing at week 4. A significantly negative correlation to the reduction in wound size for IL-6 and a similar trend for IL-8 were reported [67]. Other investigators could not identify an association between baseline levels of IL-1β, IL-1Ra, IL-6 and TNF-α and subsequent healing [50].

Two studies stood out [36,64]. Stacey et al. followed patients for 13 weeks, aiming to identify a predictive biomarker for the healing. They screened for 26 cytokines/receptors. The ulcer area, if reduced over two-week intervals, at the middle time-point of three weeks, was classified as healing, while in cases of increasing, the wound was judged as non-healing. GM-CSF and IL-16 were elevated in non-healing VLU, while IL-6 and PDGF-BB were elevated in healing VLUs by univariate analysis. GM-CSF was the sole cytokine remaining significant after multivariable logistic regression [64]. In contrast, Beidler et al. [36] reported elevated baseline tissue levels of GM-CSF in addition to IFN-γ, IL-1α, IL-1β and IL12p40 in rapid healers (>40% reduction in ulcer area in 4 weeks).

Wound fluid levels of VEGF correlated well with those in tissue, although VEGF levels were exclusively elevated in wound fluid in non-healing VLUs [44]. In one study, cytokine levels in wound fluid were higher than in plasma for FGF-2, G-CSF, GM-CSF, IFNγ, IL-1β, IL-1Ra, IL-6, IL-8, IL-10, IL-12, IL-17, TNF-α and VEGF [54].

### 3.5. Influence of Background Factors on Cytokine Levels

#### 3.5.1. Compression Therapy

Two studies assessed the effect of compression therapy in compression naïve patients [36,57]. Compression reduced the cytokine levels in tissue [36] and in serum [57]. Beidler et al. [36] investigated the effect of a multilayer bandage (Profore or Profore lite) over four weeks in 29 VLU patients. Biopsies at baseline were compared with biopsies at the end of the treatment, acquiring paired data. A reduction in G-CSF, GM-CSF, IFN-γ, IL-1α, IL-1β, IL-6, IL-8, IL12p40 and TNF-α was found following compression. The only cytokine increasing with compression was TGF-β1, with a similar trend for IL-10 (*p* = 0.076). Murphy et al. [57] compared baseline serum levels of TNF-α and VEGF in eight non-healing VLU patients, with levels after 4–6 weeks of four-layer compression “when the ulcers exhibited healing”. Both cytokines were significantly elevated in the non-healing phase and reduced with compression.

#### 3.5.2. Other Factors

Gohel et al. [48] and Wiegand et al. [67] studied the effect of wound size and age. They found higher levels of FGF-2 [48], IL-1β and IL-8 in larger VLUs, and a positive correlation between TNF-α levels and wound age [67]. Notably, VEGF levels decreased with the age of the patient [48].

Elevated IL-1, IL-6, IL-8, TNF-α and VEGF levels were reported in VLUs with high bacterial bioburden in three studies [67,68,69]; two of these were not among the included 28 studies [68,69]. Other investigators found no differences in the bacterial bioburden between healing and non-healing VLUs, despite decreases in the IL-1, IL-6 and TNF-α levels with healing [33].

Concomitant arterial disease in mixed venous-arterial ulcers did not influence the levels of EGF, FGF-2, IL-1α, IL-1β, IL-6, PDGF, TGF-β1 and TNF-α in one study [33].

The site of the biopsy influenced cytokine levels. EGF, FGF-2, IL-1α, IL-6, PDGF-A, TGF-β1, TNF-α, TNF-RI and VEGF levels were increased in perilesional skin in non-healing vs. healing VLUs; the differences were less apparent at the ulcer edge [65]. In one study, FGF-2 was increased in the ulcer edge but not in the wound bed of healing vs. non-healing VLUs [53].

### 3.6. Bioactivity of Cytokines and Cytokine Receptors

The bioactivity of cytokines was investigated in three studies [33,37,66]. The study by Harris et al. [37] failed to identify differences in the bioactivities of IL-1α, IL-1β and IL-6 in non-healing vs. healing VLUs. The authors noted that most of the IL-1α determined by ELISA assay was biologically inactive. Trengove et al. [33] reported significantly increased levels of bioactive IL-1 and IL-6 in non-healing compared to healing VLUs. Wallace et al. [66] found that total but not bioactive TNF-α was significantly elevated in non-healing vs. healing VLUs.

The expression of the cytokine receptors PDGFRα [43], TNF-RI [64,65,66], TNF-RII [64,65,66] and VEGF-R1 [44] has been studied. The gene expression of *PDGFRα* was upregulated in non-healing vs. healing VLUs [43]. Tian et al. [65] found elevated TNF-RI by immunohistochemical analysis of keratinocytes in intact skin but not in wound edge of non-healing vs. healing VLUs. sTNF-RII levels were increased in wound fluid from non-healing vs. healing VLUs [66].

### 3.7. Quality Assessment of Included Studies

All but four studies [36,48,57,61] were judged as having an overall high risk of bias, as shown in Figure 2. This was mainly due to the high risk of bias in the domains of study participation (D1), confounding (D5) and statistical analysis and reporting (D6). A high risk of selection bias was judged for studies investigating cytokines in a subgroup of VLU patients, e.g., patients admitted to the hospital for split-skin grafting. Other studies provided insufficient confirmation of the diagnosis of a VLU or provided a lack of baseline data regarding the predefined key characteristics: age, ulcer duration, ulcer size, ABPI and infection status. Few studies investigated, accounted for or adjusted for confounders. Only two studies [33,64] performed a multivariable analysis but did not include any or all of the predefined confounders. Statistical reporting and analysis were often subject to a high risk of bias, with an insufficient presentation of negative results, inappropriate selection of statistical methods and selective reporting of results. No study was corrected for multiple comparisons.

Studies lacking objective evaluation of the outcome were judged as high risk of bias unless a secondary evaluator was involved in the judgment. Quantitative methods such as ELISA and multiplex assays were regarded as low risk of bias for the cytokine measurement, despite not describing the blinding of the evaluator. Semi-quantitative measurements of immunohistochemistry were regarded as having a risk of bias if the evaluation was not performed by two blinded assessors. The risk of bias with motivations for each study can be obtained by request. We deemed pooling data in the form of a meta-analysis as inappropriate due to the overall high risk of bias and substantial clinical and methodological heterogeneity in study outcomes, study designs, time-points of biomarker collection and the specimen investigated. There was a general lack of absolute levels of cytokines, diverse data presentation and lack of report of non-significant results in many of the included papers.

## 4. Discussion

This is the first systematic review investigating the association between endogenous cytokines, including growth factors, and VLU healing. We identified possible candidate pro-inflammatory cytokines; IL-1α, IL-6, IL-8, TNF-α and VEGF levels were elevated in the non-healing phase, while they decreased when VLUs were judged to be healing. Compression therapy, which initiates VLU healing, appeared to reduce the levels of these cytokines [36,57]. In contrast, TGF-β1 levels increased with healing. Except for VEGF and TGF-β1, there was no convincing evidence for differences in growth factor levels between non-healing and healing VLUs [71,72]. The overall results are summarized in Figure 3.

The definition of healing varied among the studies; this was one of the reasons that we could not conduct a meta-analysis. Kantor and Margolis found that the wound area reduction in percentage from baseline to 4 weeks was predictive of complete VLU healing at 24 weeks [73]. Cardinal et al. confirmed the value of this 4-week metric [74]. The European Wound Management Association acknowledges a 50% wound area reduction as a valid surrogate endpoint for complete wound closure without any fixed time interval [8]. One-third of the included studies in this systematic review had follow-up times of three weeks or shorter or assessed healing clinically and subjectively. Furthermore, the effect of important confounders such as ulcer size and duration were rarely accounted for [42,48,67,75]. In addition, the effect of the bacterial bioburden on cytokine levels cannot be neglected [33,67,68,69]; increased IL-1, IL-6, IL-8, TNF-α and VEGF levels have been found in VLUs with high bacterial bioburden [67,68,69]. Cytokines influence the expression of each other. IL-1β is one of the most potent inducers of IFN-γ, IL-1Ra, IL-6, TNF-α and VEGF in multiple cell types [76]. This highlights the need for multivariable analysis, which all but two studies undertook in this review [33,64]. 

The impact of the ischemic component in mixed venous-arterial leg ulcers on cytokine levels requires further research. Results from isolated studies indicate that hypoxia upregulates the VEGF expression [77] and that arterial ulcers have higher IL-6 wound fluid levels compared to those with pure venous etiology [78]. On the other hand, Trengove et al. [33] found no effect of ischemia on the levels of cytokines in mixed venous-arterial ulcers. The included studies accepting an ABPI < 0.8 as a selection criterion seldom stated the number of patients with concomitant ischemia, and thus, it was not possible to carry out a subgroup analysis on the effect of ischemia on cytokine levels in mixed venous-arterial ulcers.

Wound fluid was the most common tissue subjected to analysis despite some inherent disadvantages [48,64]. Insufficient procured wound fluid volumes from small or dry wounds are especially problematic in longitudinal studies [48]. The duration of wound fluid collection influences cytokine levels [79]. Supposedly, local cytokine variations within a wound are not accounted for when measured in wound fluid [53,65].

In most studies, the total cytokine concentrations were measured without acknowledging their different molecular forms [80], and it is important to underscore that a high concentration of a specific cytokine does not necessarily translate into high biological activity [33,37,66]. Furthermore, the chosen method of analysis and sources of analytical reagents, e.g., antibodies and assay kits, may explain differences in reported cytokine levels [19,66]. 

The dynamic evolution of cytokines during the healing process has been closely monitored in acute wounds in humans. Vogt et al. identified different expression patterns during wound healing of partial-thickness wounds through daily measurements of a selected panel of cytokines. EGF, FGF-2, PDGF-AB and TGF-β1 levels were high early on and then decreased, while the levels of IL-1α and TGF-β2 increased with time and coincided with the epithelialization of these superficial wounds [24]. It is notable that IL-1α was associated with healing in that study [24], whereas elevated IL-α was associated with non-healing VLUs in our review. This suggests that frequent sampling (e.g., weekly) may be necessary to catch cytokine deviations between non-healing and healing VLUs.

Multiple cell lineages contribute to the cocktail of cytokines in the wound. The specific palette of elevated cytokines (IL-1α, IL-6, IL-8, TNF-α and VEGF) suggests that primarily the innate immune system is involved in the pathogenesis of non-healing VLUs. These cytokines are produced by polymorphonuclear leukocytes and macrophages [81,82,83,84,85,86] but also by keratinocytes and fibroblasts [87,88,89]. In contrast, cytokines associated with the adaptive immunity, such as the Th2 cytokines IL-4, IL-5, IL-10 and IL-13, and Th17 cytokines IL-17 and IL-22 [90] did not differ significantly between non-healing and healing VLUs in our study. 

From a theoretical perspective, a general down-regulation of inflammatory cytokines with healing is not surprising but rather expected. If a wound is to be epithelialized, excessive inflammation should subside at some time-point. The crucial question is whether elevated cytokine levels are the cause or effect of a non-healing VLU. Beidler et al. [36] reported increased levels of several inflammatory cytokines in biopsies at baseline for rapid healing VLUs compared to slow healing VLUs. In corroboration of this finding is that patients with chronic venous insufficiency (without VLUs) had significantly lower levels of GM-CSF, IL-4, IL-6, IL-7 and TNF-α in the blood compared to healthy controls, indicating that patients with chronic venous insufficiency have a subdued inflammatory state [91]. An early study by Herrick et al. supports the existence of an initially increased inflammation in normal VLU healing [92]. Two weeks of compression therapy of chronic VLUs was accompanied by a surge of inflammatory cells in the wounds, including polymorphonuclear leukocytes and macrophages, and a switch to a healing phenotype. Subsequently, the inflammatory infiltrate decreased with further healing [92].

Few studies have investigated the effect of immunosuppression on leg ulcer healing [93,94,95,96]. Diverse immunosuppressive treatments are normally known to affect acute wound healing negatively [97,98] and may be the reason why few investigators have dared to manipulate inflammation by classical immunosuppressants in VLUs. The increased risk of infections in acute wounds by certain immunomodulants [98,99] likely has a deterrent effect. The success of TNF-inhibitors in the treatment of psoriasis, rheumatoid arthritis [100] and pyoderma gangrenosum [101] is well-known, but the positive effects in VLUs have only been reported in case series [95,96]. Enhanced wound healing by blockage of TNF-α has been reported in a mice model with excessive inflammation [102], which indicates that under certain circumstances TNF-α may inhibit wound healing. 

Some limitations of this review require attention. We performed a comprehensive literature search with a sensitive search string. We did not use too-general terms, such as “chronic wound”, as it would have yielded an excessive number of irrelevant reports perhaps at the expense of missed studies that did not clearly state that VLUs were investigated in the title or abstract. Additionally, we might have missed some synonyms for the included cytokines. Restricting ourselves to English publications is a further limitation. We noticed that the time-point of cytokine measurement and its association with the outcome were sometimes unclearly described, and data extraction in this review was based on our best attempt and interpretation. The included studies were characterized by moderate to high risk of bias, and many were underpowered. The different methodologies applied made comparisons among studies difficult and prevented us from conducting a formal meta-analysis and subgroup analyses.

## 5. Conclusions

We conclude that non-healing VLUs may be associated with high levels of IL-1α, IL-6, IL-8, TNF-α and VEGF, which seemed to be reduced with healing. Conversely, TGF-β1 levels were reported to increase with healing. These results must be seen in the light of the overall moderate to high risk of bias and the many non-significant, inconsistent and conflicting results. The lack of consensus regarding the definition of healing further complicates the picture. Whether the relationship between cytokine levels and ulcer healing is consequential or casual is yet to be determined. Further longitudinal studies with head-to-head comparisons between healing and non-healing VLU with repetitive measurements of cytokines could uncover the natural history and dynamics of these cytokines. Results from such studies will increase the likelihood of identifying possible candidates of diagnostic, prognostic or therapeutic biomarkers for VLU healing.

## Figures and Tables

**Table 1 ijms-23-06526-t001:** Included studies (*n* = 28) arranged in alphabetic order.

Author	N ^1^	Study Type	Follow-Up Time	Specimen	Analysis	Cytokine ^2^
Beidler 2009 [36]	29	Cohort	4 weeks	Biopsy	Luminex, ELISA	G-CSF, GM-CSF, IFN-γ, IL-1α, IL-1β, IL-1Ra, IL-2, IL-4, IL-5, IL-6, IL-7, IL-8, IL-10, IL12p40, IL-12p70, IL-13, IL-15, IL-17, TGF-β1, TNF-α
Charles 2008 [43]	10	Cross-sectional	0	Biopsy	cDNA microarray	HB-EGF, PDGFRα, S100A7
Drinkwater 2003 [44]	35	Cohort	~1 year	WF, biopsy	ELISA, RT-PCR	VEGF-121, VEGF-165, VEGF-189, VEGF-RI, VEGF-RII
Escandon 2012 [45]	10	Clinical trial ^3^	4 weeks	Biopsy	RT-PCR	IL-1α, IL-6, IL-8, IL-10, IL-11, TNF-α, VEGF
Filkor 2016 [46]	69	Cohort	4 weeks	PBMC	qRT-PCR	IL-1α, IL-8, IL-10, TNF-α
Fivenson 1997 [47]	14	Cohort	8 weeks	WF	ELISA	IL-8, IL-10
Gohel 2008 [48]	80	Cohort	5 weeks	WF, serum	ELISA	FGF-2, IL-1β, TGF-β1, TNF-α, VEGF
Grandi 2018 [49]	19	Clinical trial ^4^	~3 weeks	Biopsy	IHC	TGF-β, TNF-α
Harris 1995 [37]	18	Cross-sectional	0	WF	ELISA, bioassay	FGF-2, GM-CSF, IL-1α, IL-1α bio, IL-1β bio, IL-6 bio, PDGF-AB
He 1997 [50]	10	Experimental ^5^	200 min	Serum	ELISA	IL-1β, IL-1Ra, IL-6, TNF-α
Hodde 2020 [51]	12	Clinical trial ^6^	12 weeks	WF	Luminex, ELISA	GM-CSF, IFN-γ, IL-1β, IL-2, IL-4, IL-5, IL-8, IL-10, IL-12(p70), IL-13, TGF-β1, TNF-α
Krejner 2017 [52]	19	Cohort, retrospective	-4 weeks	Serum	ELISA	IL-6, IL-8, TNF
Lagattolla 1995 [53]	19	Cohort ^7^	6 months	Biopsy	ELISA	FGF-2, PDGF-AB, TGF-β1,
Ligi 2016 [54]	34	Cross-sectional, cohort	During admission	WF	Luminex	FGF-2, G-CSF, GM-CSF, IFN-γ, IL-1β, IL-1Ra, IL-2, IL-4, IL-5, IL-6, IL-7, IL-8, IL-9, IL-10, IL-12(p70), IL-13, IL-15, IL-17, PDGF-BB, TNF-α, VEGF
Ligi 2017 [55]	30	Cross-sectional, cohort	0	WF	Luminex	TGF-β1, TGF-β2, TGF-β3
McQuilling 2021 [56]	15	Clinical trial ^8^	12 weeks	WF	Multiplex MAP arrays	EGF, FGF-2, G-CSF, GM-CSF, IFN-γ, IL-1α, IL-1β, IL-1Ra, IL-2, IL-3, IL-4, IL-5, IL-6, IL-7, IL-8, IL-9, IL-10, IL-12p40, IL-12p70, IL-13, IL-15, IL-17, PDGF-AA, PDGF-BB, TGF-α, TGF-β1, TGF-β2, TGF-β3, TNF-α, TNF-β, VEGF
Murphy 2002 [57]	8	Cohort	12 weeks	Serum	ELISA	TNF-α, VEGF
Mwaura 2006 [58]	40	Cohort	8 weeks	WF, biopsy	ELISA, IHC	PDGF-AA
Pukstad 2010 [59]	8	Cohort	8 weeks	WF	Antibody array, bioassay	IL-1α, IL-1β, IL-6R, IL-8, TNF-α, sTNF-RI, sTNF-RII
Sadler 2012 [60]	20	Clinical trial ^9^	4 weeks	WF	ELISA	TNF-α
Senet 2003 [61]	15	RCT ^10^	16 weeks	WF	ELISA	IL-8, KGF, VEGF
Serra 2013 [62]	60	RCT ^11^	~13 months	WF, plasma	ELISA	VEGF
Serra 2015 [63]	64	RCT ^9^	~13 months	WF, plasma	ELISA	VEGF
Stacey 2019 [64]	42	Cohort	13 weeks	WF	Multiplex ELISA	G-CSF, GM-CSF, IFN-γ, IL-1α, IL-1β, IL-1Ra, IL-2, IL-4, IL-5, IL-6, IL-7, IL-8, IL-10, IL-11, IL-12p40, IL-12p70, IL-13, IL-15, IL-16, IL-17, M-CSF, PDGF-BB, TNF-α, TNF-β, TNF-RI, TNF-RII
Tian 2003 [65]	21	Cohort	2 weeks	Biopsy	IHC	EGF, FGF-2, IL-1α, IL-6, PDGF-A, TGF-β1, TNF-α, TNF-RI, VEGF
Trengove 2000 [33]	26	Cohort	2 weeks	WF	ELISA, bioassay	EGF, FGF-2, IL-1α, IL-1β, IL-1 bio, IL-6, IL-6 bio, TGF-β1, TNF-α, PDGF
Wallace 1998 [66]	21	Cohort	2 weeks	WF	ELISA, bioassay	TNF-α, TNF-α bio, sTNF-RI, sTNF-RII
Wiegand 2017 [67]	42	RCT ^12^	4 weeks	WF, biopsy	Luminex, IHC	IL-1β, IL-6, IL-8, IL-10, TGF-β, TNF-α

^1^ Total number of enrolled patients. ^2^ Cytokines fulfilling the inclusion criteria of the review. ^3^ Ultrasound. ^4^ Aminolevulinic acid photodynamic therapy. ^5^ Reperfusion injury. ^6^ Small-intestine submucosa. ^7^ Retrospective or prospective study. ^8^ Amniotic membrane. ^9^ Doxycycline. ^10^ Autologous platelet gel vs. placebo. ^11^ Minocycline vs. control. ^12^ Ultrasound vs. control. ELISA, enzyme-linked immunosorbent assay; IHC, immunohistochemistry; qRT-PCR, real-time quantitative reverse transcription polymerase chain reaction; RCT, randomized controlled trial; WF, wound fluid.

**Table 2 ijms-23-06526-t002:** Impact of cytokines on VLU healing *.

Cytokine	Non-Healing VLUs	vs.	Healing VLUs
EGF	↑ Biopsy [65]		↑ WF [56]
FGF-2	↑ Biopsy [65]		↑ Biopsy [53]
GM-CSF	↑ WF [54,64]		↑ Biopsy [36] ^1^
HB-EGF			↑ Biopsy [43]
IFN-γ			↑ Biopsy [36] ^1^
IL-1α	↑ WF [33,56], ↑ Biopsy [45,65], ↑ Blood [46]		↑ Biopsy [36] ^1^
IL-1β	↑ WF [33,54]		↑ Biopsy [36] ^1^
IL-1 bio ^3^	↑ WF [33]		
IL-1Ra	↑ WF [56]		↑ Biopsy [36] ^2^
IL-2	↑ WF [56]		
IL-3	↑ WF [56]		
IL-6	↑ WF [67], ↑ Biopsy [45,65]		↑ WF [64] ^4^
IL-6 bio ^3^	↑ WF [33]		
IL-7	↑ WF [56]		
IL-8	↑ WF [51,54,56,61], ↑ Biopsy [45], ↑ Blood [46]		↑ WF [47]
IL-9	↑ WF [56]		
IL-10	↑ WF [54]		
IL-11	↑ Biopsy [45]		
IL-12p40			↑ Biopsy [36] ^1^
IL-12p70	↑ WF [54]		
IL-16	↑ WF [64] ^4^		
PDGF-AA	↑ Biopsy [65]		↑ WF [58], ↑ Biopsy [58]
PDGF-BB			↑ WF [54,64] ^4^
PDGFRα	↑ Biopsy [43]		
S100A7			↑ Biopsy [43]
TGF-β1	↑ Biopsy [65]		↑ WF [48,51], ↑ Biopsy [49,53]
TGF-β3	↑ WF [55]		
TNF-α	↑ WF [33,51,66], ↑ Biopsy [45,65], ↑ Blood [57] ^2^		
TNF-β	↑ WF [56]		
TNF-RI	↑ Biopsy [65]		
TNF-RII	↑ WF [66]		
VEGF	↑ WF [44,54,62,63], ↑ Biopsy [45,65], ↑ Blood [57,62,63] ^2^		

*↑ Increase (*p* < 0.10) of cytokine in wound fluid (WF), biopsy or blood. ^1^ Before initiation of compression at baseline. ^2^ After 4 weeks of compression. ^3^ Bioactivity. ^4^ Univariate analysis.

**Figure 1 ijms-23-06526-f001:**
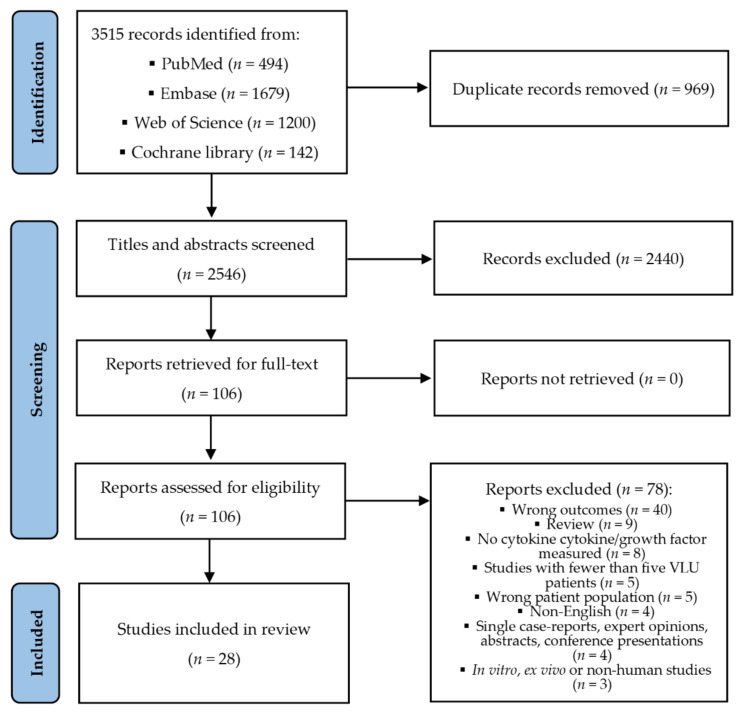
PRISMA flow diagram [39].

**Figure 2 ijms-23-06526-f002:**
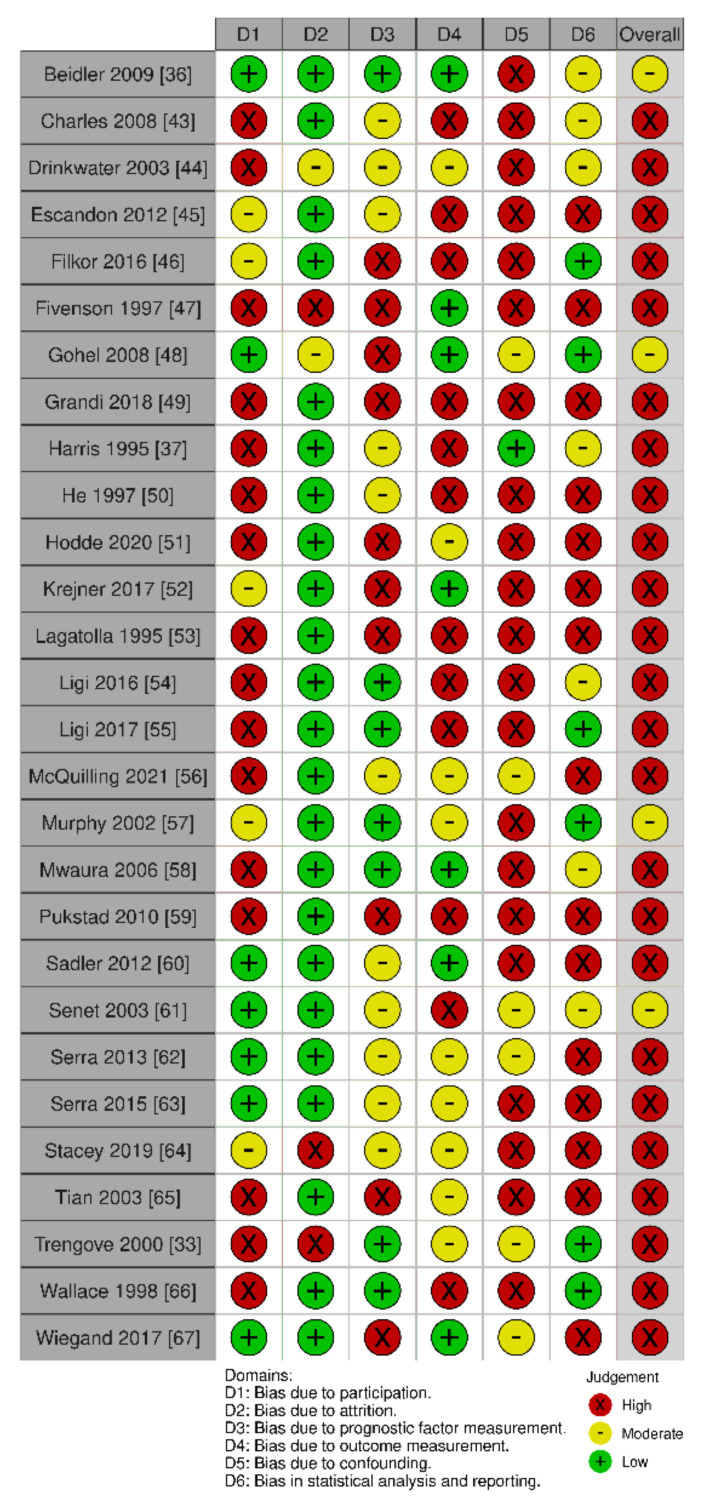
Quality assessment of the 28 included studies (risk of bias). The individual studies were judged in six domains (D1–D6), and each domain was judged as high, moderate or low risk of bias. An overall rating took all domains into account. The traffic-light plot was created with a tool by McGuinness et al. [70].

**Figure 3 ijms-23-06526-f003:**
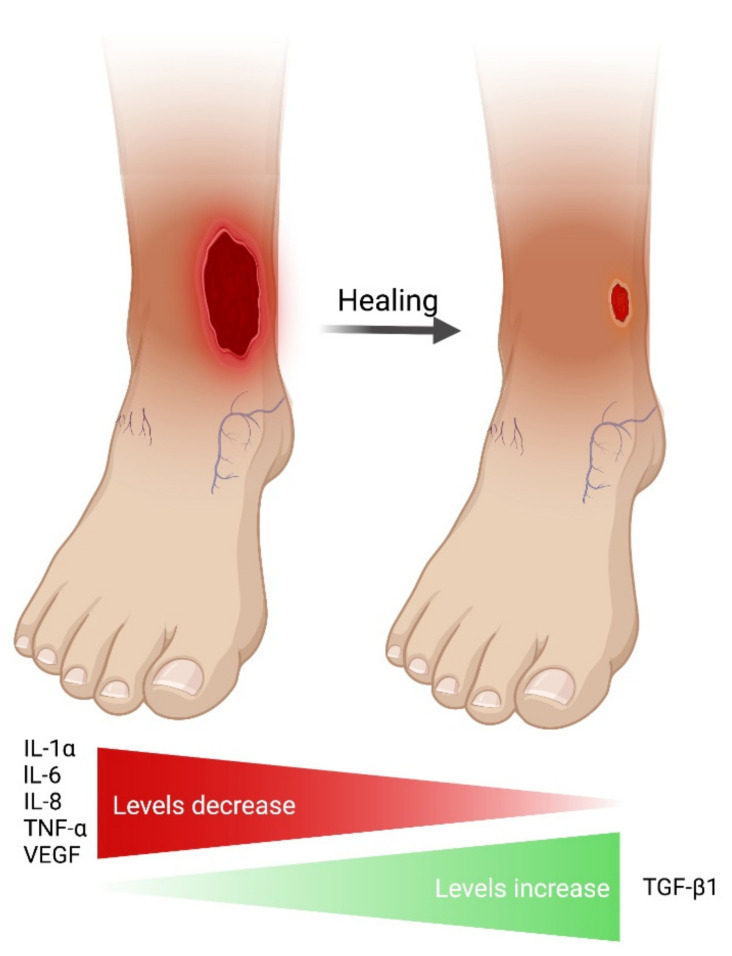
Elevated cytokines (*p* < 0.10) shown in at least three studies in this review.

## Data Availability

The data presented in this study are available in the article or the Appendix A. The risk of bias assessment for each article included can be achieved by request.

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
