# Peer review of "Cytokines and Venous Leg Ulcer Healing—A Systematic Review"

_ijms, 2022, doi:10.3390/ijms23126526_

Round 1
Reviewer 1 Report
The manuscript is well-written and presents good methodology with extensive literature search. The manuscript also clearly describes the limitations, both of the articles reviewed and of this review. However, I have some remarks/questions.
1. In the introduction section, the majority of the literature is quite old and I would suggest to add some more recent references, such as Guest JF, Fuller GW, Vowden P. BMJ Open 2020 Dec.
2. The authors should explain more clearly why also mixed venous-arterial ulcers were included, as the pathophysiology differs from venous leg ulcers. Did the cytokine profiles differ in these ulcers? This is an important question the reader wants to understand.
Author Response
Dear reviewer,
Thank you for your valuable feed-back on our manuscript.
- Thank you for making us aware of this important reference (Guest et al. 2020). It has now been added to the introduction.
- Thank you for raising this important topic. We evaluated ischemia as an important confounder, similarly to infection, immunosuppression and diabetes. Therefore, we did not exclude these studies. Studies accepting an ABPI <0.8 as a selection criterion rarely stated the number of patients included with concomitant ischemia as a baseline factor. Therefore, it was not possible to assess the number of patients with ischemia and thus a subgroup analysis was not plausible. We hope we have raised the issue of confounders in the results, discussion and limitations more clearly now. Future studies would need to investigate the impact of ischemia more thouroughly. Interestingly, Trengove et al. (2000) did not identify that ischemia impacted the production of cytokines:
“When compared statistically to those ulcers with purely venous disease, there was no significant difference in the levels of cytokines or in the rates of cell proliferation. This suggests that the presence of arterial disease in these patients does not influence the production of cytokines in the wound.”
The lack of reporting and accountability of ABPI is reflected by our risk of bias judgment (these studies would not be judged as having a low risk of bias). We hope our answer and changes in the manuscript are satisfying.
We have added the fowllowing to the Discussion section dealing with this this tpic:
"The impact of the ischemic component in mixed venous-arterial leg ulcers on cytokine levels requires further research. Results from isolated studies indicate that hypoxia upregulates the VEGF expression [77] and that arterial ulcers have higher IL-6 wound fluid levels compared to those with pure venous etiology [78]. On the other hand, Trengove et al. [33] found no effect of ischemia on the levels of cytokines in mixed venous-arterial ulcers. The included studies accepting an ABPI < 0.8 as a selection criterion seldom stated the number of patients with concomitant ischemia and thus it was not possible to carry out a subgroup analysis on the effect of ischemia on cytokine levels in mixed venous-arterial ulcers."
Reviewer 2 Report
It is a well-written paper. I suggest accepting it in its present form. Please check the file attached.

Author Response
Dear Reviewer,
Thank you very much for your valuable comments. We acknowledge that the methods used for cytokine assessment are diverse, as well as the specimen used for analysis (wound fluid, biopsies and serum/plasma). Specifically, we have evaluated that a subgroup analysis of the heterogenous clinical and methodological data would be too challenging and potentially misleading. We have added some discussion on the effect on ischemia in mixed venous-arterial ulcers on cytokine levels. The included studies accepting an ABPI < 0.8 as a selection criterion rarely stated the number of patients included with concomitant ischemia. Therefore, it was not possible to assess the number of patients with ischemia and thus a subgroup analysis was not plausible. We hope these issues are addressed properly and accurately in the Discussion:
“The impact of the ischemic component in mixed venous-arterial leg ulcers on cytokine levels requires further research. Results from isolated studies indicate that hypoxia upregulates the VEGF expression [77] and that arterial ulcers have higher IL-6 wound fluid levels compared to those with pure venous etiology [78]. On the other hand, Trengove et al. [33] did not identify that ischemia impacted the levels of cytokines in mixed venous-arterial ulcers.
Wound fluid was the most common tissue subjected to analysis despite some inherent disadvantages [48, 64]. Insufficient procured wound fluid volumes from small or dry wounds is especially problematic in longitudinal studies [48]. The duration of wound fluid collection influences cytokine levels [73]. Supposedly, local cytokine variations within a wound are not accounted for when measured in wound fluid [53, 65].
In most studies the total cytokine concentrations were measured without acknowledging their different molecular forms and it is important to underscore that a high concentration of a specific cytokine does not necessarily translate into high biological activity [37, 66]. Furthermore, the chosen method of analysis and sources of analytical reagents, e.g., antibodies and assay kits, may explain differences in reported cytokine levels [66]."